# The Mechanism of Oxymatrine Targeting miR-27a-3p/PPAR-γ Signaling Pathway through m6A Modification to Regulate the Influence on Hemangioma Stem Cells on Propranolol Resistance

**DOI:** 10.3390/cancers15215213

**Published:** 2023-10-30

**Authors:** Yuxin Dai, Mingke Qiu, Shenglai Zhang, Jingyu Peng, Xin Hou, Jie Liu, Feifei Li, Jingmin Ou

**Affiliations:** 1Department of Intervention and Vascular Surgery, XinHua Hospital, Shanghai Jiaotong University School of Medicine, Shanghai 200025, China; daiyuxin139@sjtu.edu.cn (Y.D.); qiumingke@xinhuamed.com.cn (M.Q.); pengjingyu8563@xinhuamed.com.cn (J.P.); 2Department of General Surgery, Shigatse People’s Hospital, Shigatse 857000, China; 3Department of General Surgery, XinHua Hospital, Shanghai Jiaotong University School of Medicine, Shanghai 200025, China; zhangshenglai@xinhuamed.com.cn; 4Department of Intervention and Vascular Surgery, Chongming Branch of Xinhua Hospital, Shanghai Jiaotong University School of Medicine, Shanghai 200025, China; 722008563@shsmu.edu.cn (X.H.); xueguanyujieru@sumhschongming.com (F.L.); 5Department of Interventional & Vascular Surgery, Shanghai Tenth People’s Hospital, Shanghai 200072, China; lj10935@easthospital.cn

**Keywords:** hemangioma, oxymatrine, propranolol, adipose differentiation, drug resistance

## Abstract

**Simple Summary:**

In this study, the researchers aimed to understand the mechanism behind the resistance of hemangioma stem cells (HemSCs) to propranolol, a commonly used drug for hemangioma treatment. They investigated the role of a specific signaling pathway involving miR-27a-3p and PPAR-γ, as well as the impact of a treatment called oxymatrine (OMT). The findings revealed that miR-27a-3p negatively controlled the peroxisome-proliferator-activated receptor γ (PPAR-γ), which contributed to the resistance of HemSCs to propranolol. OMT treatment accelerated the progression and adipocyte differentiation of HemSCs via modulating the miR-27a-3p/PPAR-γ axis, thus inhibiting their resistance to propranolol. This research sheds light on the potential of OMT as a therapeutic strategy for hemangiomas and highlights the importance of targeting the miR-27a-3p/PPAR-γ pathway. These findings may have implications for improving the effectiveness of propranolol (PPNL) treatment and advancing the understanding of hemangioma biology in the research community.

**Abstract:**

Objective: The proliferation and migration of hemangioma stem cells (HemSCs) induced apoptosis and adipose differentiation as well as increased the sensitivity of HemSCs to propranolol (PPNL). MiR-27a-3p negatively controlled the peroxisome-proliferator-activated receptor γ (PPAR-γ) level, counteracting the effect of PPAR-γ on HemSC progression and PPNL resistance. OMT accelerated HemSC progression and adipocyte differentiation via modulating the miR-27a-3p/PPAR-γ axis, inhibiting HemSC resistance to PPNL. In tumor-forming experiments, OMT exhibited a dose-dependent inhibitory effect on the volume of IH PPNL-resistant tumors, which was partially dependent on the regulation of m6A methylation transfer enzyme METTL3 and the miR-27a-3p/PPAR-γ axis, thereby inducing apoptosis. Conclusions: We conclude that OMT regulates IH and influences PPNL resistance via targeting the miR-27a-3p/PPAR-γ signaling pathway through m6A modification.

## 1. Introduction

Infantile hemangioma (IH) is a common benign vascular tumor that tends to occur in 3–10% of infants and young children [1,2]. Its pathological development can be divided into three stages: a period of rapid proliferation shortly after birth, a regression period from several months to several years after birth, and the completion period of regression in childhood [3]. Although IH can spontaneously degenerate, it may lead to disfigurement, organ dysfunction, and the death of some children, bringing varying degrees of negative psychological impacts and economic burden to patients and their families [4]. Propranolol (PPNL) is the first-line drug for IH treatment, but some infants with IH will show resistance to PPNL treatment to varying degrees [5]. However, the exact pathogenesis of IH is still unclear, and reports have confirmed that IH originates from CD133+ IH-derived stem cells (HemSCs) [6]. HemSCs, like stem cells, are pivotal for the pathological development of IH and can differentiate into hematoma-derived endothelial cells, pericytes, and adipocytes at the stage of degeneration [7,8]. This study will analyze the molecular resistance mechanism of IH and PPNL based on HemSCs, aiming to provide a reference value for the effective treatment of IH.

Oxymatrine (OMT) is a quinolizidine alkaloid isolated from the traditional Chinese herbal medicine Sophora flavescenta Aiton, which has anti-inflammatory, anti-fibrotic, antiviral, anti-cancer, and other pathological effects [9]. Current studies have shown that OMT can prevent proliferation and trigger the programmed cell death of IH cells through inhibiting hypoxia-inducible factor 1A (HIF-1A) signaling, suggesting that OMT has a potential anti-IH protective effect and may offer a novel method to treat IH [10]. N6-methyladenosine (m6A) was first identified as a modification on mammalian internal messenger RNA and acts as the most common modification on mRNAs and long noncoding (lnc) RNAs. The microRNA (miRNA)/messenger RNA (mRNA) molecular regulatory network plays a regulatory role in most biological processes, as well as a potential regulatory role for drug-resistant tumors and other specific diseases [11]. miRNAs are short single-stranded non-coding molecules that regulate cells and molecular processes via binding to target mRNA [12]. miR-27a-3p, as a member of miRNA, has been confirmed to decrease calcium deposits in vascular smooth muscle cells and improve atherosclerosis through the targeted activation of transcription factor 3 (ATF3) [13]. Other research has shown that miR-27a-3p can affect the programmed cell death, autophagy, and inflammatory response of vascular endothelial cells via mediating the lncRNA taurine up-regulated gene 1 (TUG1)/miR-27a-3p/slit directed ligand 2 (SLIT2) molecular network [14]. Moreover, miRNA-27a-3p and miRNA-222-3p are novel modulators of phosphodiesterase 3a (PDE3A) in cerebral microvascular endothelial cells (PMID: 30603956). However, PPAR-γ participates in the evolution of IH and can promote an increase in adipocytes to replace tumors with fibrous adipose tissue, thus exerting specific potential therapeutic effects on IH [15]. Moreover, miR-27a-3p has a targeted relationship with PPAR-γ, and the former can improve sevoflurane-induced cognitive impairment through the negative regulation of the latter [16]. Taken together, the miR-27a-3p/PPAR-γ molecular regulatory network may mediate the pathogenesis of IH.

Studies have shown OMT both inhibits and promotes the proliferation and cell apoptosis of breast cancer cells, respectively, which may contribute to the regulation of miRNA-140-5p (PMID: 35035706). Moreover, OMT inhibits HSC activation through down-regulating the expression of miR-195 (PMID: 31221141). However, there are few studies on the mechanism by which OMT regulates the miRNA/mRNA axis, which affects the pathologic development of IH and PPNL resistance. In this study, we attempted to corroborate the association between OMT and the miR-27a-3p/PPAR-γ axis and its anti-IH protection mechanism, thus addressing a gap in the existing literature.

## 2. Materials and Methods

### 2.1. Flow Chart

A flow chart of the full text is shown in Figure 1.

### 2.2. Ethics Statement

All patients who participated in the study willingly signed informed consent forms. The Clinical Trial Ethics Committee of Xinhua Hospital, affiliated with Shanghai Jiaotong University School of Medicine, approved all experimental procedures.

### 2.3. Obtaining and Grouping Tissue Specimens of Patients with IH

Eighty patients with infantile hemangioma were enrolled in our study, admitted to our hospital between January 2016 and January 2020. Among these patients, 30 were in the proliferative stage, 28 were in the regression stage, and 22 were in the completion stage of IH. The parents or guardians of all afflicted individuals provided their informed consent for participation in this research. Our study rigorously adhered to the principles of the Declaration of Helsinki and was fully supported by the ethical approval granted by our hospital’s ethics committee.

### 2.4. Reverse Transcription-Polymerase Chain Reaction (RT-PCR)

For total RNA extraction, 1 mL of Trizol Reagent (Shandong Weida Industrial Co., Ltd., Shanghai, China; 15596018) was added to 100 mg of tissue. Then, samples were homogenized and thoroughly ground until no visible tissue remained. The supernatant was then centrifuged at 13,000× *g* for 10 min. Next, 250 μL of trichloromethane was added, and the centrifugal tube was inverted for 15 s to mix thoroughly. It was allowed to stand for 3 min, followed by centrifugation at 13,000× *g* at 4 °C for 8 min. The resulting supernatant was transferred to a new centrifugal tube, and 0.8 times the volume of isopropyl alcohol was added. The mixture was gently inverted, placed at −20 °C for 15 min, and then subjected to centrifugation at 13,000× *g* at 4 °C for 600 s. The liquid was carefully removed, and 1.5 mL of 75% ethanol was added to clean the precipitation. The sample was centrifuged at 13,000× *g* at 4 °C for 300 s. Subsequently, 20 μL of RNA-free water was added to dissolve the RNA, and the mixture was incubated at 55 °C for 5 min.

For reverse transcription, 2 μg of RNA was added. Then, 1 μL of Oligo (dT) was added, followed by the addition of deionized water (without ribonuclease) to bring the total volume to 12 μL. This mixture was incubated at 70 °C for 5 min using a PCR instrument and quickly placed on ice to cool. To this mixture, 4 μL of 5× buffer, 2 μL of 10 mM dNTPs, 1 μL of RNA inhibitor, and 1 μL of reverse transcriptase were successively added. The PCR instrument (Shanghai Jingxin Industrial Development Co., Ltd., Shanghai, China; TLAN-96) was then set to 42 °C for 60 min. Finally, the reverse transcriptase was inactivated at 80 °C for 5 min. The contents of miR-27a-3p and PPAR-γ mRNA were detected, and the specific sequences were as follows: miR-27a-3p-forward primer: 5′-TGCGCTTCACAGTGGCTAAGT-3′, reverse primer: 5′-CCAGTGCAGGGTCCGAGGTATT-3′, PPAR-γ-forward primer: 5′-CTGGCCTCCCTGATGAATAA-3′, reverse primer: 5′-CGCAGGTTTTTGAGGAACTC-3′. Internal reference U6 forward primer: 5′-CGCTTCGGCAGCACATATAC-3′, reverse primer: 5′-AAATATGGAACGCTTCACGA-3′, internal reference GAPDH forward primer: 5′-TGTGTCCGTCGTGGATCTGA-3′, reverse primer: 5′-CCTGCTTCACCACCTTCTTGAT-3′.

### 2.5. Western Blot

Tissue samples were cleaned twice with a 0.01 M phosphoric acid buffer solution at 4 °C (PBS). Afterward, they were lysed with a mixture of lysate and protease inhibitor (99:1 ratio, SY4680, YT2336) on ice for 30 min. The lysate was then centrifuged at 12,000 r/s at 4 °C for 600 s, and the supernatant was collected. It was then mixed with sample-loading buffer and boiled for 600 s. Subsequently, SDS polyacrylamide gel electrophoresis (SDS-PAGE; Beijing Ita Biotechnology Co., Ltd., Beijing, China; SY4336) was performed, and the separated proteins were transferred onto a polyvinylidene fluoride membrane ((PVDF) Beijing Dehang Wuzhou Technology Co., Ltd., Beijing, China; UE427144). The PVDF membrane was blocked with 5% fat-free milk for 2 h, and then the primary antibodies were added and incubated overnight at 4 °C. Following this, the secondary antibodies were added and incubated at room temperature for 1 h. Detection was performed using an ECL luminescent reagent (Shanghai Chuan-Qiu Biotechnology Co., Ltd., H-E-60). The expression levels of PPAR-γ (ab45036, 1:500, Abcam, Cambridge, Britain), C/EBPβ (ab15050, 1:500, Abcam, Cambridge, Britain), C/EBPδ (ab245214, 1:1000, Abcam, Cambridge, Britain), RXRα (ab227273, 1:500, Abcam, Cambridge, Britain), RXRγ (AF9191, 1:300, Affinity, Cincinnati, USA), METTL3 (#86132, 1:1000, Cell Signaling Technology, Boston, MA, USA), METTL14 (#48699, 1:1000, Cell Signaling Technology, Boston, MA, USA), WTAP (#56501, 1:1000, Cell Signaling Technology, Boston, MA, USA), and KIAA1429 (ab246982, 1:500, Abcam, Cambridge, Britain) were analyzed.

### 2.6. Immunohistochemistry

The tissue specimens were fixed using a solution of 4% paraformaldehyde in PBS and then subjected to a 30% sucrose solution for sinking. Subsequently, sections were treated with primary antibodies and incubated overnight. After thorough cleaning, the sections were sequentially incubated to secondary antibodies and an avidin –biotin complex (ABC) of related species. The sections were then developed using the diaminobenzidine (DAB) method, mounted, dehydrated, and sealed. The results were analyzed, and differences between groups were compared.

### 2.7. Extraction and Isolation of HemSCs

The IH specimens were initially dissected into small pieces using a scalpel, followed by enzymatic digestion using collagenase (Beijing Ita Biotechnology Co., Ltd., Beijing, China; YT0601). The resulting cell suspension was filtered through a 100 μm cell filter. Subsequently, the cells were subjected to magnetic sorting using strains coated with CD133 antibodies (Shenzhen Haodi Huatuo Biotechnology Co., Ltd., Shenzhen, China; PL0401125) to isolate CD31+ cells, which were identified as HemSCs. These HemSCs were then cultured on cell culture dishes coated with fibronectin (Grenner Biotechnology (Shanghai) Co., Ltd., Shanghai, China; 628920). The culture medium used was Endothelial Cell Growth Medium-2 (EGM-2; Shanghai Weijin Biotechnology Co., Ltd., Shanghai, China; CC-3162), supplemented with 20% fetal bovine serum (FBS; Beijing Zeping Technology Co., Ltd., Beijing, China; 76294-180(17)).

### 2.8. Cell Transfection

The construction of the miR-27a-3p overexpression plasmid (miR-27a-3p) and corresponding negative control (mimic-NC), the miR-27a-3p inhibitory plasmid (intrex-miR-27a-3p) and corresponding negative control (intrex-NC), and the PPAR-γ overexpression plasmid (OE-PPAR-γ) and corresponding negative control (OE-NC) were synthesized by Shanghai Jima Biotechnology Company (Shanghai, China) and transfected into cells via Lipofectamine 2000 transfection reagent (Beijing Lambokance Technology Co., Ltd., Beijing, China; 11668027).

Different concentrations (0.0, 0.1, 0.5, 1.0, and 2.0 mg/mL) of OMT (Beijing Ita Biotechnology Co., Ltd.; YT63902) were added to the HemSCs culture medium; then, the cells were cultured in a 37 °C incubator for 48 h. Untreated cells were used without any drug or plasmid intervention.

### 2.9. Establishment of HemSC/PPNL Cell Line

Construction of PPNL-resistant HemSCs (HemSCs/PPNL) occurred via exposing HemSCs to 2μM OF PPNL (Shanghai Yiji Industrial Co., Ltd., Shanghai, China; YJ-318989). When the cells were in the logarithmic growth phase, a suspension of 5 × 10^7^ cells was prepared to start the follow-up experiment. A 10 mL cell suspension was inoculated in a culture flask for 24 h; then, PPNL was added (starting from 2 μM) and incubated for 48 h. The new solution was replaced, 5 μM PPNL was added after digestion, and then processed for 48 h. Then, the concentration of PPNL was gradually increased to 10 μM and 20 μM, and HemSCs/PPNL cells that could tolerate 20 μM of PPNL were obtained. HemSCs/PPNL cells treated with OMT were treated as the PPNL + OMT group, and HemSCs/PPNL cells treated with the same volume buffer (Beijing Huizhi Heyuan Biotechnology Co., Ltd., Beijing, China; 0111261) were treated as the PPNL + Buffer group.

### 2.10. CCK-8 Assay

The cells were incubated at 37 °C with 100 μL cell suspension in a 96-well plate. Cell proliferation was checked at 0, 24, 48, and 72 h. Cells in each group were added with 10 μL of CCK8 solution (Shanghai Fuheng Biotechnology Co., Ltd., Shanghai, China; C200-10) and incubated at 37 °C for 4 h. The absorption of every sample was identified at 450 nm via a spectrophotometer (Shanghai Zhenghuang Trading Co., Ltd., Shanghai, China; UH5700).

### 2.11. Transwell Assay

A Transwell chamber without matrix gel was used to examine cell migration. HemSCs were seeded into 6-well plates, 5 × 10^4^ cells were supplemented to the upper chamber, while RPMI-1640 medium with 10% FBS was supplemented to the lower chamber. Migrating cells treated with crystal violet (SND-B2572) were analyzed via an inverted microscope (Shanghai Yuyan Scientific Instrument Co., Ltd. Shanghai, China).

### 2.12. Apoptosis Assay

After digestion and centrifugation via trypsin (Beijing Ita Biotechnology Co., Ltd.; YT1598), the cells were washed via iced PBS and subjected to resuspension with 1× binding buffer. Annexin V-FITC/PI kit (Shanghai Jingke Chemical Technology Co., Ltd., Shanghai, China; AD10-2) was used for the determination. The procedure was strictly followed as per the supplier’s instructions. Finally, the apoptotic cells were detected via a flow cell technique (Beijing Beijiamei Biotechnology Co., Ltd. Beijing, China), and the apoptosis rate was calculated.

### 2.13. Dual-Luciferase Reporter Gene Assay

The PPAR-γ 3’UTR of miR-27a-3p and PPAR-γ binding spot mutation was constructed, and mutant-type and wild-type PPAR-γ (PPAR-γ-MUT and PPAR-γ-WT, respectively) were inserted into the pGL3-BASIC plasmid to construct luciferase reporter gene vector. Subsequently, PPAR-γ-MUT and PPAR-γ-WT were cotransfected into HemSCs with miR-27a-3p and mimic-NC, respectively. The cells were collected 48 h later, the luciferase activity in HemSCs was detected according to the luciferase activity detection kit (Shanghai Caiyou industrial Co., Ltd., Shanghai, China; c-8303), and the relative fluorescence intensity was calculated.

### 2.14. MeRIP-RT-PCR

A Magna RIP RNA binding protein immune precipitation tool (Guangzhou Boxin Biotechnology Co., Ltd., Guangzhou, China; Bes5101) was used to determine RIP. The cells were statically transfected with METTL3-overexpressed plasmid (OE-MeTTL3) and its control (OE-NC) were lysed with RIP lysis buffer. M6A antibody (Shanghai Jingfeng Biotechnology Co., Ltd., Shanghai, China; TF10757R) and magnetic beads were added to the RNA solution, slowly shaken, and incubated at 4 °C overnight. The RNA was eluted from the magnetic beads, and the RNA from the IP was collected. TaqMan^®^ Pri-miRNA Assay Kit (DxT-4441127) was used to detect the pri-miRNA content in all RNA.

### 2.15. Oil Red O Staining

HemSCs were inoculated into 6-well plates at 1 × 10^6^ cells/mL per well, and given different treatments. After 48 h, the cell medium was replaced with adipocyte differentiation medium (Beijing Qingyuanhao Biotechnology Co., Ltd., Beijing, China; LL-0059). The cells were incubated at 37 °C for 10 days; then, oil red O (Chuzhou Shinoda Biotechnology Co., Ltd., Chuzhou, China; SND1237) cells were stained for 30 min and photographed with a Nikon Eclipse E800 microscope (Shanghai Tucson Vision Technology Co., Ltd. Shanghai, China).

### 2.16. Tumor Xenograft Experiment

The HemSCs/PPNL cell density was adjusted to 5 × 10^7^/mL, and 200 μL was inoculated in both armpits of Balb/C nude mice (Hunan Silke Jingda Experimental Animal Co., Ltd. Changsha, China), with 10 mice in each group. When the tumor was visible to the naked eye, 25 mg/kg and 50 mg/kg of OMT were intraperitoneally injected into the OMT-L and OMT-H groups, respectively, and the injection of an equal dose of PBS was given to the Ctrl group. On days 0, 3, 7, 14, and 21 of OMT treatment, the tumor tissues of mice were taken out and photographed, and the volume was recorded. The expressions of METTL3, miR-27a-3p, and PPAR-γ in the tumorous tissues were identified via RT-PCR. Animal experiments were evaluated and accepted by the Ethical Board for Animals.

### 2.17. Statistics

The data were presented as the mean and standard deviation (SD), and group comparisons were assessed using Student’s *t*-test or one-way ANOVA for normally distributed data. Mann–Whitney or Kruskal–Wallis tests were employed for comparing groups with non-normally distributed data. Pearson correlation analyses were used to evaluate the association between miR-27A-3p and PPAR-γ expression. All statistical analyses were conducted using SPSS 23.0, with statistical significance set at *p* < 0.05.

## 3. Results

### 3.1. MiR-27a-3p Was Up-Regulated and PPAR-γ Was Down-Regulated in IH

We collected tissue samples from 80 patients diagnosed with IH, distributed as follows: 30 in the proliferation stage, 28 in the regression stage, and 22 in the completion stage. It was observed that the expression level of miR-27A-3p was highest in patients at the proliferation stage, followed by those in the regression and completion stages, as illustrated in Figure 2A. Conversely, as shown in Figure 2B and Appendix A, the protein level of PPAR-γ was lowest in the proliferation stage, followed by the regression and completion stages. Immunohistochemistry analysis further confirmed that PPAR-γ expression was the lowest in individuals in the proliferation stage, significantly lower than in those in the regression and completion stages, as depicted in Figure 2C.

To delve deeper into the relationship between miR-27a-3p and PPAR-γ, we conducted a correlation analysis, which revealed a negative correlation between miR-27a-3p and PPAR-γ. Notably, this negative correlation was prominent during the proliferation period (r = −0.583, *p* < 0.05, Figure 2D) and regression stage (r = −0.513, *p* < 0.05, Figure 2D), while no statistically significant differences in this negative association were observed among patients after regression (r = −0.221, *p* > 0.05, Figure 2D).

### 3.2. Effects of miR-27A-3p on the Biological Function of HemSCs

Our analysis of miR-27a-3p expression confirmed its abnormal dysregulation in IH, suggesting that miR-27a-3p may play a role in mediating the deterioration process of IH. For in vitro studies, HemSCs were isolated from the tissues of patients with proliferating IH, and an inhibitory sequence for miR-27a-3p was constructed and transfected into HemSCs. RT-PCR validation demonstrated a significant reduction in miR-27a-3p expression in HemSCs transfected with the inhibitory miR-27a-3p sequence (Figure 3A). Functional experiments revealed that the proliferation (Figure 3B) and migration (Figure 3C,D) of HemSCs were significantly impeded following the suppression of miR-27a-3p expression. Additionally, the level of apoptosis showed a substantial increase (Figure 3E,F). These findings indicate that miR-27a-3p may indeed contribute to the regulation of HemSCs behavior and, consequently, the progression of IH.

### 3.3. Effects of PPAR-γ on the Biological Function of HemSCs

Likewise, we analyzed the impact of PPAR-γ on HemSCs in vitro. The protein expression level of PPAR-γ in HemSCs was notably increased through the use of the PPAR-γ overexpression vector, as confirmed via Western blot analysis (Figure 4A and Appendix A). The results indicated that the overexpression of PPAR-γ significantly reduced the proliferation of HemSCs (Figure 4B) and impaired their migration capability (Figure 4C,D). Additionally, a significant induction of apoptosis was observed (Figure 4E,F).

### 3.4. MiR-27a-3p Exerts a Negative Regulatory Effect on PPAR-γ

We validated the association between miR-27a-3p and PPAR-γ using various biological tools and identified potential targeting sites for both, as shown in Figure 5A. Following the successful construction of the miR-27a-3p overexpression sequence (Figure 5B), we transfected it into HemSCs containing PPAR-γ-WT or PPAR-γ-MUT for a dual-luciferase activity test. Notably, we observed a significant reduction in fluorescence activity when miR-27a-3p was overexpressed, but this reduction occurred only in cells containing PPAR-γ-MUT and not in those with PPAR-γ-WT (Figure 5C). Subsequent validation experiments further confirmed that miR-27a-3p overexpression led to a substantial decrease in both the mRNA and protein levels of PPAR-γ (Figure 5D–F and Appendix A).

### 3.5. Effects of miR-27A-3p/PPAR-γ Signaling Pathway on Biological Functions of HemSCs

Furthermore, we evaluated the role of the miR-27a-3p/PPAR-γ signaling pathway in the biological functions of HemSCs. We observed that miR-27a-3p could counteract the inhibitory effects of PPAR-γ overexpression on HemSCs’ proliferation (Figure 6A) and migration (Figure 6B,C), as well as the promoting effect of PPAR-γ overexpression on HemSCs’ apoptosis (Figure 6D,E). Based on the findings from these comprehensive studies, we can conclude that miR-27a-3p can target the expression of PPAR-γ, consequently exerting a significant influence on the biological functions of HemSCs.

### 3.6. Effect of miR-27A-3p/PPAR-γ Signaling Pathway on Adipose Differentiation of HemSCs

In the pathological progression of infantile hemangioma (IH), adipose differentiation signifies a pathological improvement, and IH gradually transitions from the proliferative stage to the degenerative stage. Additionally, PPAR-γ plays a crucial role in fat formation and lipid storage. Therefore, we also investigated the roles of the miR-27a-3p/PPAR-γ signaling pathway in the adipose differentiation of HemSCs. We assessed this through oil red O staining and observed that the suppression of miR-27a-3p or the overexpression of PPAR-γ could enhance lipid accumulation in HemSCs (Figure 7A,B). Furthermore, the protein levels of C/EBPβ, C/EBPδ, RXRα, and RXRγ were also found to increase to varying degrees (Figure 7C,D and Appendix A).

### 3.7. Effects of miR-27A-3p/PPAR-γ Signaling Pathway on HemSCs/PPNL-Resistant Cells

In addition, our team tested the roles of the miR-27A-3p/PPAR-γ signaling pathway in HemSCs/PPNL-resistant cells. It was discovered that the expression of miR-27a-3p was remarkably higher in HemSCs/PPNL cells than in HemSCs (Figure 8A). Under the intervention of different concentrations of PPNL, the proliferation ability of both cell types was reduced in a dose-dependent manner with PPNL, and the proliferation level of HemSCs/PPNL cells was significantly higher than that of HemSCs (Figure 8B). In terms of apoptosis, the apoptosis level of HemSCs was significantly increased under PPNL stimulation, while the apoptosis level of HemSCs/PPNL cells was not significantly changed (Figure 8C,D). These results suggest that the drug resistance model of HemSCs was successfully constructed, and HemSCs/PPNL-resistant cells exhibited more significant tumor malignancy and PPNL resistance. However, the proliferation level and IC50 value of HemSCs/PPNL cells exposed to PPNL were significantly down-regulated in a dose-dependent manner upon down-regulation of miR-27a-3p or up-regulation of PPAR-γ, compared with those exposed to PPNL alone. However, the co-expression of miR-27a-3p and FGFR1 offset the above effects and, finally, showed no significant effect on cell proliferation and IC50 values (Figure 8E,F).

### 3.8. Effects of OMT on Adipose Differentiation and PPNL Resistance of HemSCs

For the sake of exploring the anti-IH protective causal link of OMT, we studied lipid differentiation and PPNL resistance. Oil red O dyeing revealed that OMT significantly elevated the lipidic accumulation of HemSCs (Figure 9A) and promoted the protein level of adipose-differentiation-related proteins (Figure 9B and Appendix A). In addition, the anti-IH mechanism of OMT can cooperate with the increasing concentration of PPNL to effectively inhibit cell proliferation (Figure 9C), reduce IC50 (Figure 9D), and induce cell apoptosis (Figure 9E,F).

### 3.9. OMT Enhances the Sensitivity of HemSCs to PPNL via Regulating miR-27a-3p/PPAR-γ Axis through Modifying m6A

To further understand the underlying mechanism of OMT’s anti-IH protective effect, we studied its relationship with m6A, miR-27a-3p, and PPAR-γ. The data revealed that OMT significantly inhibited the protein level of m6A methyltransferase METTL3 in HemSCs, but had no significant effect on METTL14, WTAP, and KIAA1429 protein levels (Figure 10A,B and Appendix A). RIP analysis revealed that the over-expression of METTL3 remarkably increased the level of pri-miR-27a-3p modified via m6A (Figure 10C). In addition, we also found that OMT exhibited dose-dependent suppression of miR-27a-3p (Figure 10D) and dose-dependent promotion of PPAR-γ mRNA or protein levels (Figure 10E–G and Appendix A).

### 3.10. OMT Regulation of miR-27A-3p/PPAR-γ Axis Facilitates Apoptosis In Vivo to Inhibit PPNL Resistance

To further analyze the in vivo protective mechanism of OMT against IH, we inoculated mice with HemSCs/PPNL cells and injected mice with a low concentration of OMT (OMT-L) and a high concentration of OMT (OMT-H) intravenously when tumors were visible. It was found that tumor size and tumor volume were significantly reduced in mice injected with OMT-L and significantly reduced in mice treated with OMT-H (Figure 11A,B). OMT-L also significantly inhibited METTL3 and miR-27a-3p levels and promoted PPAR-γ expression more significantly under the OMT-H intervention (Figure 11C). In addition, OMT-L also significantly induced the expression of Bax, Caspase 3, Caspase 9, and other pro-apoptotic proteins, and omT-H induced their expression more significantly (Figure 11D,E and Appendix A).

## 4. Discussion

IH tends to occur in females, Caucasians, low-weight premature infants, and multiple births, and its potential complications involve permanent disfigurement, scarring, dysopsia, hyperemia, cardiac failure, and even mortality in severe cases [17]. OMT is a quinolizidine alkaloid extracted from the traditional Chinese herbal medicine Sophora flavescenta Aiton, known for its various pathological effects, including its anti-inflammatory, anti-fibrotic, antiviral, and anti-cancer properties, among others [9]. Current research has indicated that OMT can inhibit the proliferation of IH cells via suppressing hypoxia-inducible factor 1A (HIF-1A) signaling, as well as inducing programmed cell death, suggesting its potential protective effect against IH and offering a novel approach for IH treatment [10]. N6-methyladenosine (m6A) is a modification found in mammalian internal messenger RNA, playing regulatory roles in numerous biological processes and potentially contributing to drug resistance in tumors and specific diseases [11]. The miRNA/mRNA molecular regulatory network regulates various biological processes via binding to target mRNA, indicating potential roles in drug-resistant tumors and specific diseases [11]. Additionally, PPAR-γ participates in IH progression and can promote an increase in adipocytes to replace tumors with fibrous adipose tissue, potentially offering specific therapeutic effects in IH [15]. In summary, the miRNA/PPAR-γ molecular regulatory network may mediate the pathogenesis of IH. However, research on the mechanism by which OMT regulates the miRNA/mRNA axis affecting the pathologic development of IH and PPNL resistance remains limited. In this study, we aimed to both verify the association between OMT and the miR-27a-3p/PPAR-γ axis and explore its protective mechanism against IH.

Although traditional Chinese medicine anticancer drugs have gained popularity in treating tumors, their anti-tumor mechanisms remain to be clarified. Recent research has revealed that miRNA is closely associated with the bioactive anti-tumor effect of traditional Chinese medicine and may exert either anti-cancer or carcinogenic effects [18]. Our clinical outcomes unveil that miR-27a-3p might exert a carcinogenic effect on the development of IH, while PPAR-γ has a tumor suppressive effect, and the two may have antagonistic effects on the proliferation and regression stages of IH. Past research has revealed that miR-27a-3p functions as a carcinogen in renal cell carcinoma, and high levels of it have been related to unfavorable prognoses in patients [19]. miR-27a-3p also exerts carcinogenic effects in gastric cancer and can affect the cycle and apoptosis of gastric carcinoma cells via targeting B cell translocation gene 2 (BTG2) [20]. In a study by ABD-Elbaset et al. [21], PPAR-γ was shown to play an anticancer role in hepatocellular carcinoma via mediating antioxidant mechanisms. In addition, PPAR-γ can also play an anticancer role in osteosarcoma via inducing tumor cell apoptosis and inhibiting tumor growth [22]. In terms of the mechanisms of action, the suppression of miR-27a-3p or over-expression of PPAR-γ exhibit specific therapeutic effects on IH cells in vitro, and both can exert protective effects via inhibiting growth, metastasis, and inducing apoptosis of HemSCs in vitro. We identified a targeted association between miR-27A-3p and PPAR-γ through biological tool evaluation, and the former can negatively regulate the mRNA and protein contents of PPAR-γ. Furthermore, the miR-27a-3p/PPAR-γ axis exerts a regulatory effect on the pathological progression of HemSCs.

PPAR-γ pertains to the nuclear hormone receptor family of transcription factors, which also include PPAR-α and PPAR-β [23]. PPAR-γ is a transcription factor that regulates fat formation and lipid storage and can promote the expression of adipogenesis-related genes via forming a DNA heterodimer with a vitamin A-X receptor-α (RXR-α) [24,25]. Studies have shown that the PPAR-γ signaling pathway can mediate the adipogenic differentiation of mesenchymal stem cells into adipocytes, and the accumulation of adipocytes during IH pathology indicates the transition from proliferation to degeneration [26]. Herein, both the inhibition of miR-27a-3p and the up-regulation of PPAR-γ promoted the adipogenic differentiation of HemSCs, and the mechanism may be associated with increased levels of PPAR-γ, C/EBPβ, C/EBPδ, RXRα, RXRγ, and other adipogenic proteins. We also observed that OMT also promoted these adipogenic proteins and significantly promoted the adipogenic differentiation of HemSCs. However, PPNL resistance is a critical treatment difficulty for patients with IH [27]. It is known that miR-187-3p can enhance the sensitivity of HemSCs to PPNL via targeting the adhesive protein-loading factor (NIPBL) [28]. Our research unveiled that the miR-27a-3p/PPAR-γ signaling pathway can regulate the PPNL resistance of drug-resistant HemSCs, and the up-regulation of miR-27a-3p or the down-regulation of FGFR1 can seriously inhibit the malignant proliferation of cells and improve the sensitivity of drug-resistant HemSCs to PPNL. Moreover, OMT has a similar function. Su et al. [29] reported that OMT could be combined with vincristine to inhibit the drug resistance of HCT-8/VCR cells, the mechanism of which is related to their regulation of autophagy activity and the activation of the TLR4 signal. Liang et al. [30] demonstrated that OMT could reverse 5-fluorouracil resistance in colon cancer cells via inhibiting the epithelial–mesenchymal transformation and nuclear factor κB (NF-κB) signaling. These studies all suggest that OMT plays a therapeutic role in tumor drug resistance and can improve the sensitivity of tumor cells to chemotherapy through some mechanism.

In terms of molecular mechanisms, our study also confirmed that OMT regulates the miR-27a-3p/PPAR-γ axis, thereby inhibiting PPNL resistance in HemSCs via modifying m6A through the down-regulation of METTL3 expression. Among them, m6A denotes the methylation modification on the sixth nitrogen atom of adenine, which can regulate the generation of non-coding RNAs, including miRNA, and can be catalyzed via several m6A methyltransferases (METTL3, METTL14, WTAP, and KIAA1429) [31]. Therefore, m6A can also be extensively involved in the pathological process of tumors via mediating miRNA expression [32]. These are key proteins involved in RNA m6A methylation. METTL3 and METTL14 constitute the core of the m6A methylation complex responsible for adding methyl groups to RNA molecules. WTAP serves as an auxiliary factor in this complex, and KIAA1429, also known as VIRMA, plays a significant role in RNA methylation. For example, METTL3 can enhance the expression level of miR-1246 via facilitating the maturation of pri-miR-1246 and, finally, induce malignant metastasis of colon cancer [33]. Our in vivo results suggest that OMT regulates the miR-27a-3p/PPAR-γ axis and promotes apoptosis in vivo via increasing the expression of pro-apoptotic proteins to inhibit PPNL resistance, which ultimately leads to the inhibition of IH tumor growth. The novelty of this study is that OMT’s anti-IH protective mechanism was confirmed from the aspects of cell function, adipose differentiation, PPNL resistance, and in vivo studies, which are related to the regulation of METTL3 expression of m6A and the miR-27a-3p/PPAR-γ axis. However, there is room for improvement in future studies. Firstly, we can increase the analysis of miR-27a-3p upstream of long non-coding RNA to supplement the molecular regulatory network further. Secondly, we can also look for molecular pathways related to OMT and analyze whether it has regulatory effects on certain molecular pathways. Finally, we can also explore the effect of OMT on angiogenesis. We will gradually improve the research in relation to the above points.

## 5. Conclusions

To the best of our knowledge, our study provides the first evidence that miR-27a-3p is abnormally up-regulated in IH proliferating tissues and HemSCs and that the down-regulation of miR-27A-3p expression can inhibit the proliferation and migration as well as induce apoptosis of HemSCs, promote adipose differentiation, and increase PPNL sensitivity. Moreover, the molecular protective mechanism of OMT in IH is related to miR-27a-3p. We also proposed for the first time that OMT regulates the pathological progression of HemSCs and increases PPNL sensitivity via targeting miR-27a-3p/PPAR-γ signaling pathway through m6A modification, providing a new strategy for the treatment of patients with IH.

## Figures and Tables

**Figure 1 cancers-15-05213-f001:**
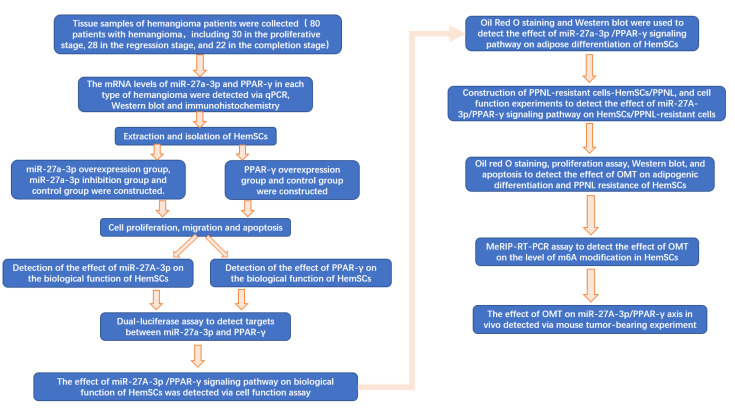
Flow chart of full text.

**Figure 2 cancers-15-05213-f002:**
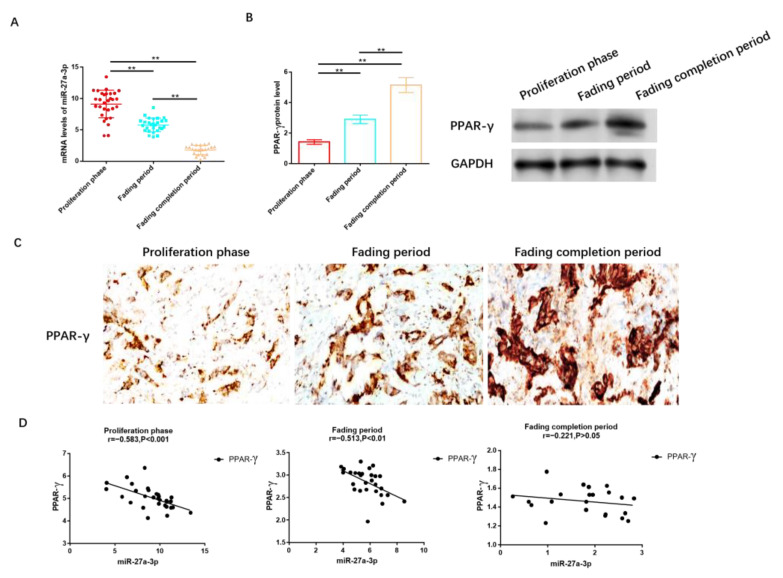
MiR-27a-3p was up-regulated in IH, while PPAR-γ was down-regulated. (**A**) RT-PCR was utilized to assess the expression of miR-27a-3p during the phases of proliferation, regression, and completion in IH. (**B**) The protein levels of PPAR-γ in the proliferation, regression, and completion stages of IH were determined using Western blot analysis. (**C**) IHC was employed to measure the expression level of PPAR-γ in tissues from the IH proliferation, IH regression, and IH regression completion stages. (**D**) Pearson correlation analysis was conducted to examine the relationship between miR-27a-3p and PPAR-γ during the IH proliferation, IH regression, and IH regression completion stages. ** *p* < 0.01. The uncropped bolts are shown in Appendix A.

**Figure 3 cancers-15-05213-f003:**
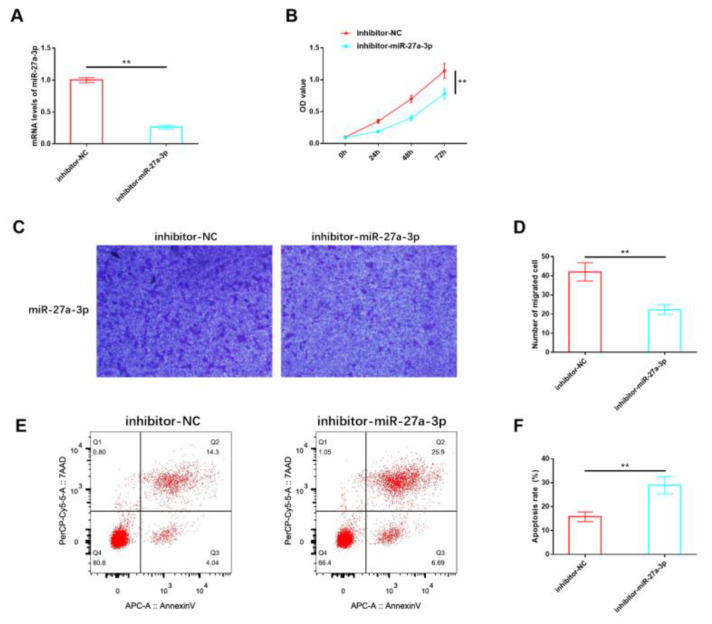
Effects of miR-27A-3p on the biological functions of HemSCs. (**A**) The transfection efficiency of the interactive miR-27a-3p was confirmed through RT-PCR. (**B**) The impact of miR-27A-3p on HemSCs proliferation was assessed using the CCK-8 assay. (**C**) HemSCs’ migration was evaluated using the Transwell method. (**D**) The corresponding statistical analysis is shown. (**E**) HemSCs’ apoptosis was detected using flow cytometry. (**F**) The associated statistical analysis is presented. ** *p* < 0.01.

**Figure 4 cancers-15-05213-f004:**
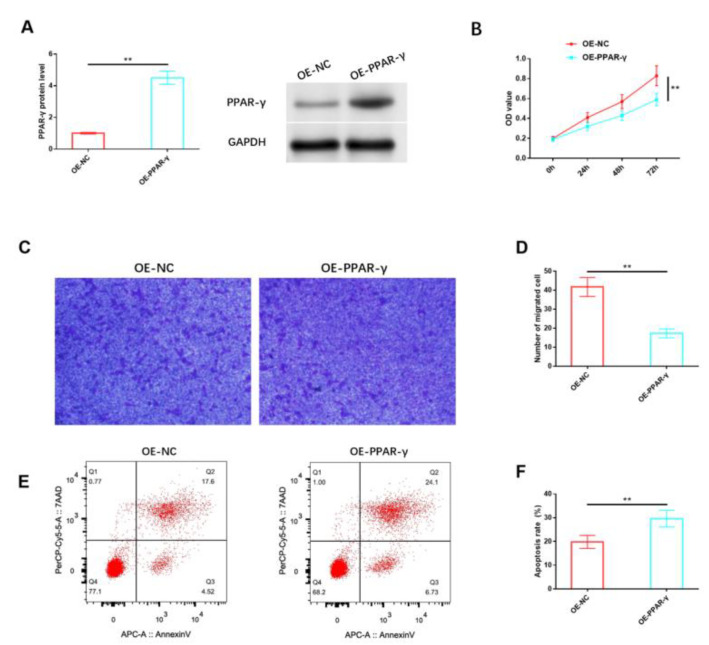
Effects of PPAR-γ on the biological functions of HemSCs. (**A**) The transfection efficiency of OE-PPAR-γ was assessed via Western blot analysis. (**B**) The impact of OE-PPAR-γ on HemSCs’ proliferation was evaluated using the CCK-8 assay. (**C**) HemSCs’ migration was determined using the Transwell assay. (**D**) The corresponding statistical analysis is presented. (**E**) HemSCs’ apoptosis was detected using flow cytometry. (**F**) The associated statistical analysis is depicted. ** *p* < 0.01. The uncropped bolts are shown in Appendix A.

**Figure 5 cancers-15-05213-f005:**
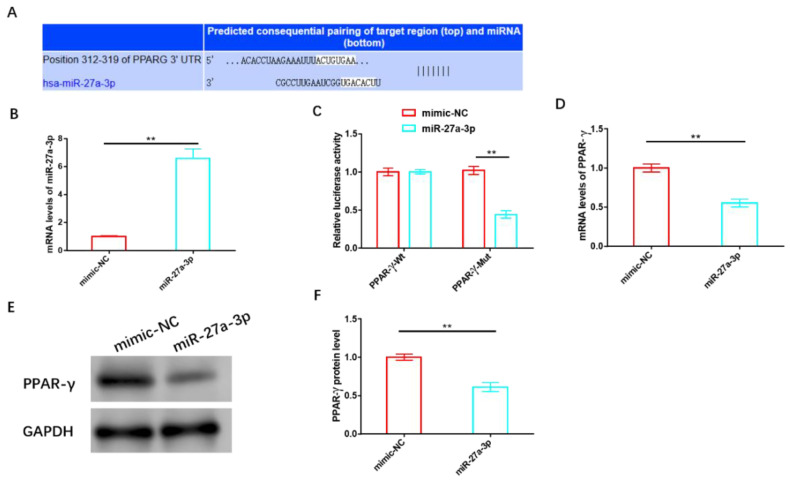
MiR-27a-3p exerts a negative regulatory effect on PPAR-γ. (**A**) Targeting sites of miR-27a-3p and PPAR-γ. (**B**) The transfection efficiency of miR-27a-3p was assessed using RT-PCR. (**C**) A dual-luciferase activity assay was performed to confirm the targeting relationship between miR-27a-3p and PPAR-γ. (**D**) RT-PCR analysis and (**E**) Western blot analysis were employed to examine the effects of miR-27a-3p on the expression and protein levels of PPAR-γ. ** *p* < 0.01. The uncropped bolts are shown in Appendix A.

**Figure 6 cancers-15-05213-f006:**
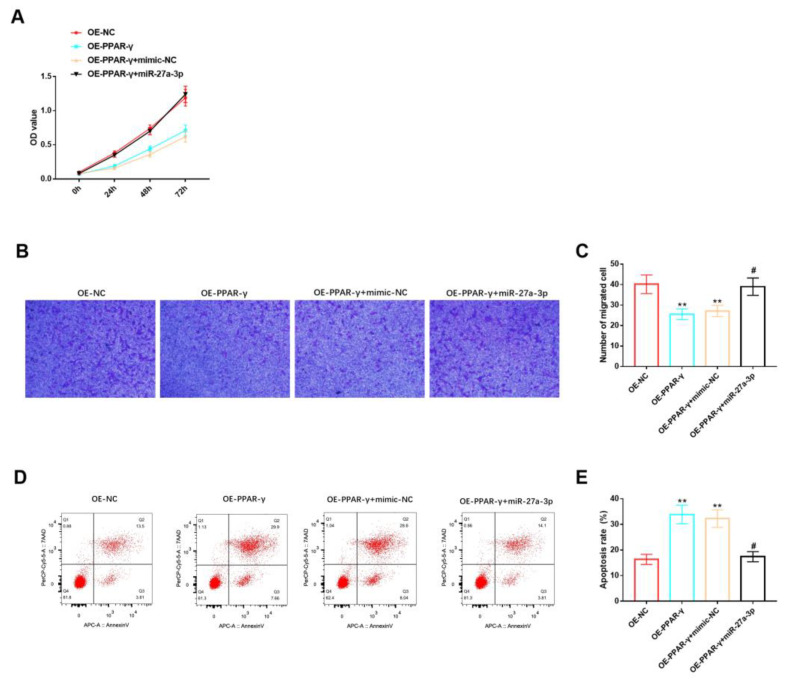
Effects of miR-27A-3p/PPAR-γ signaling pathway on biological functions of HemSCs. (**A**) The CCK-8 assay was employed to assess the impact of the miR-27a-3p/PPAR-γ signaling pathway on HemSCs’ proliferation. (**B**) HemSCs’ migration was examined using the Transwell method, and the results are presented in (**C**). (**D**) HemSCs’ apoptosis was evaluated via flow cytometry, and the corresponding statistical analysis is shown in (**E**). Comparing the experimental group with OE-NC, ** *p* < 0.01, and comparing the experimental group with OE-PPAR-γ + mimic-NC, # *p* < 0.05.

**Figure 7 cancers-15-05213-f007:**
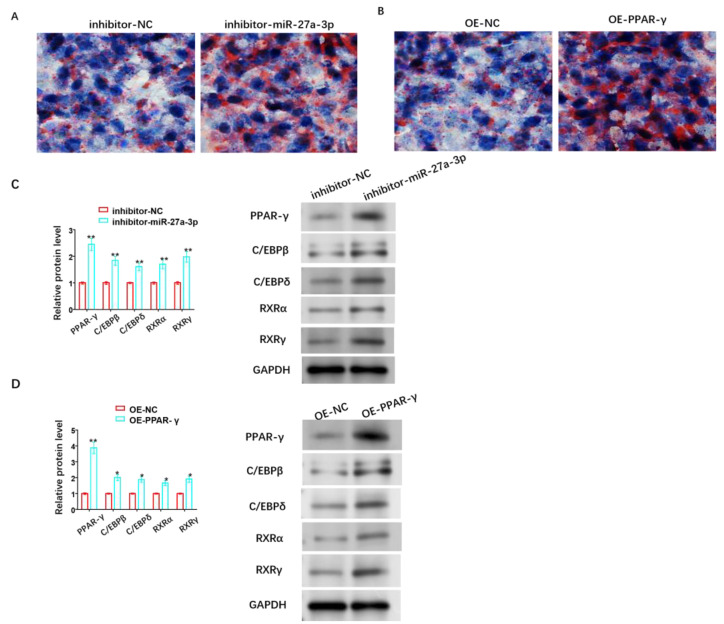
Role of miR-27A-3p/PPAR-γ signaling pathway in adipose differentiation of HemSCs. (**A**) Oil red O staining was performed to assess the impact of miR-27a-3p inhibitor. (**B**) The role of OE-PPAR-γ in the adipose differentiation of HemSCs. (**C**) Western blot analysis was used to detect the effect of miR-27a-3p inhibitor. (**D**) OE-PPAR-γ and its effect on the expression of proteins related to adipogenesis in HemSCs, including PPAR-γ, C/EBPβ, C/EBPδ, RXRα, and RXRγ. * *p* < 0.05, ** *p* < 0.01. The uncropped bolts are shown in Appendix A.

**Figure 8 cancers-15-05213-f008:**
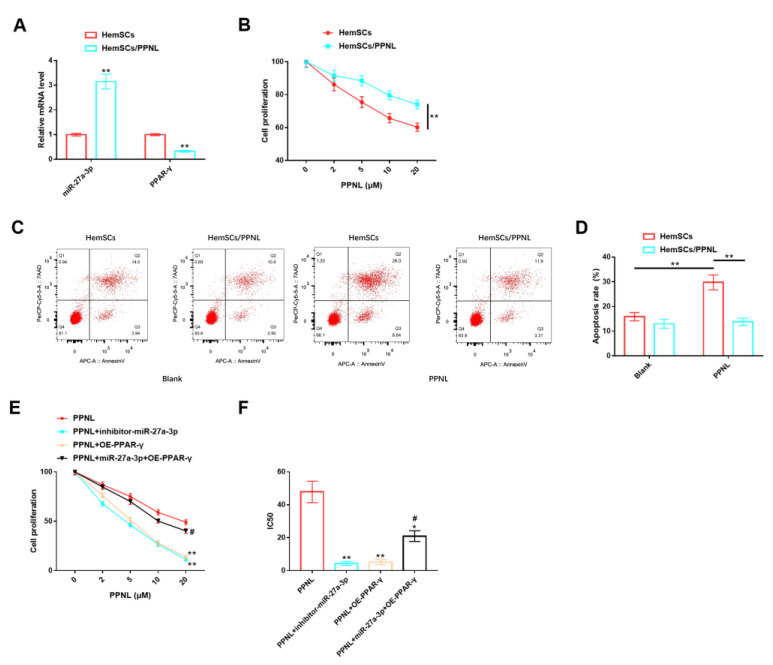
Role of miR-27A-3p/PPAR-γ signaling pathway in HemSCs/PPNL-resistant cells. (**A**): RT-PCR detection of the expression levels of miR-27a-3p and PPAR-γ in HemSCs/PPNL-resistant cells and HemSC parent cells. (**B**): CCK-8 method detection of the effects of different PPNL concentrations on the proliferation of HemSCs/PPNL-resistant cells and HemSC parent cells. (**C**): Flow cytometric detection of the apoptosis rate of HemCs/PPNL drug-resistant cells and HemSC parent cells. (**D**): Statistical graph. (**E**): CCK-8 method to analyze cell proliferation. (**F**): IC50 after PPNL treatment. In contrast to PPNL, * *p* < 0.05, ** *p* < 0.01; in contrast to PPNL + OE-PPAR-γ, # *p* < 0.05.

**Figure 9 cancers-15-05213-f009:**
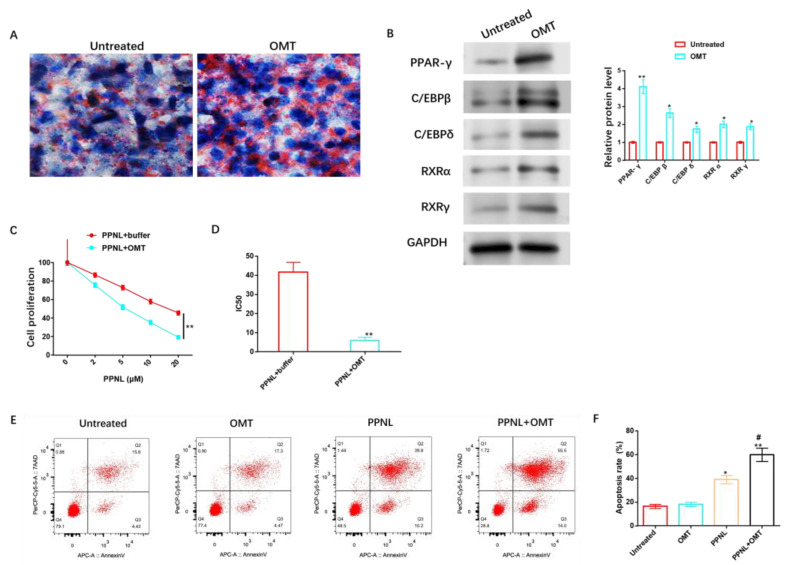
Effects of OMT on adipose differentiation and PPNL resistance of HemSCs. (**A**): Oil red O dyeing evaluates the role of OMT in the fat differentiation of HemSCs. (**B**): WB detection of the effect of OMT on HemSCs fat-differentiation-related proteins. (**C**): CCK-8 method identification of the role of OMT in the proliferative ability of HemSCs/PPNL-resistant cells. (**D**): IC50 value. (**E**): Flow cytometry detection of the effect of OMT on the apoptosis rate of HemSCs/PPNL drug-resistant cells. (**F**): Statistical graph. Compared with untreated or PPNL + buffer or between the two groups, * *p* < 0.05, ** *p* < 0.01; in contrast to PPNL, # *p* < 0.05. The uncropped bolts are shown in Appendix A.

**Figure 10 cancers-15-05213-f010:**
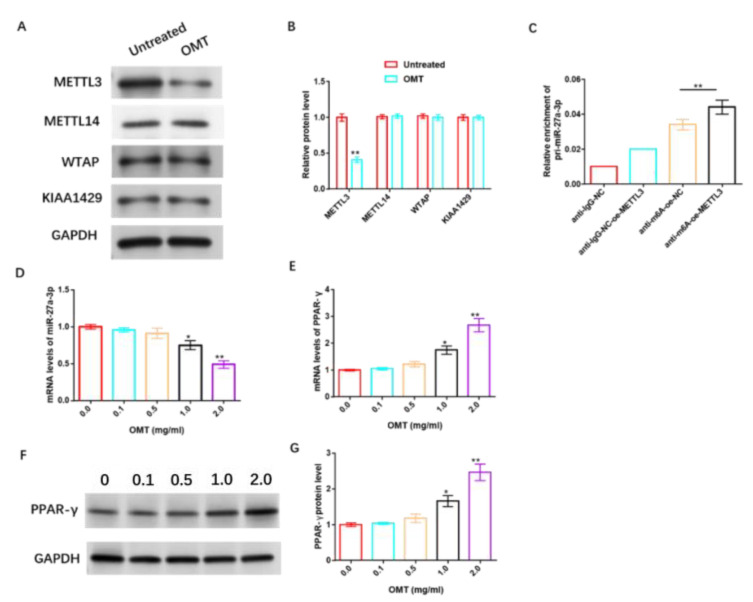
OMT enhances the sensitivity of HemSCs to PPNL via modulating miR-27a-3p/PPAR-γ axis through modifying m6A. (**A**): WB detection of the effect of OMT treatment on the protein levels of HemSCs/PPNL drug-resistant cells METL3, METL14, WTAP, and KIAA1429. (**B**): Statistical graph. (**C**): MeRIP-RT-PCR detection and analysis of the effect of METL3 overexpression on the level of m6A-modified pri-miR-27a-3p. (**D**): RT-PCR detection of the effect of OMT on HemSC miR-27a-3p. (**E**): PPAR-γ expression. (**F**): WB detection of the effect of OMT on HemSC PPAR-γ protein level. (**G**): Statistical graph. Compared with untreated or 0.0, * *p* < 0.05, ** *p* < 0.01. The uncropped bolts are shown in Appendix A.

**Figure 11 cancers-15-05213-f011:**
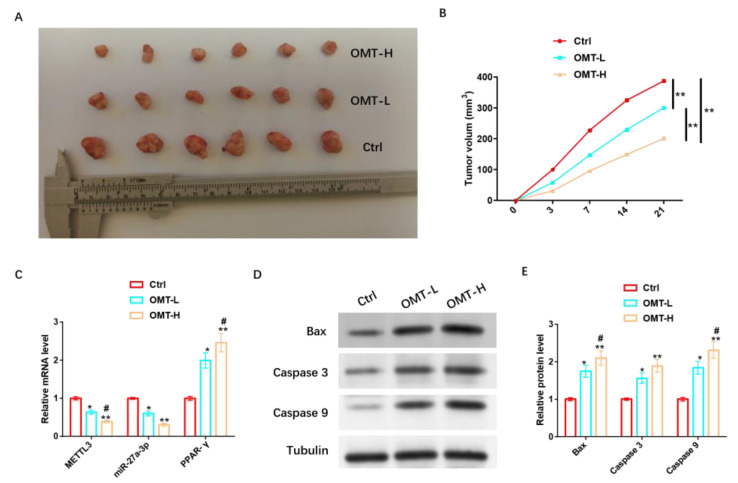
OMT regulation of miR-27a-3p/PPAR-γ axis promotes apoptosis in vivo to inhibit PPNL resistance. (**A**): Tumor formation of nude mice under different treatments. (**B**): Volume of each group. (**C**): RT-PCR detection of METL3, miR-27a-3p, PPAR-γ levels under different treatments. (**D**): WB detection of apoptosis-related proteins under different treatments; the protein expression of Bax, Caspase 3, and Caspase 9; and their (**E**) statistical graphs. Compared with Ctrl or between the two groups, * *p* < 0.05, ** *p* < 0.01; in contrast to OMT-L, # *p* < 0.05. The uncropped bolts are shown in Appendix A.

## Data Availability

The data presented in this study are available upon request from the corresponding author.

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
