# Peer review of "The Mechanism of Oxymatrine Targeting miR-27a-3p/PPAR-γ Signaling Pathway through m6A Modification to Regulate the Influence on Hemangioma Stem Cells on Propranolol Resistance"

_cancers, 2023, doi:10.3390/cancers15215213_

Round 1
Reviewer 1 Report
Dai and colleagues report “The mechanism of OMT targeting miR-27a-3p /PPAR-γ signaling pathway through m6A modification to regulate hemangioma stem cells' influence on propranolol resistance”
A major problem with the writing is that it is impossible to understand the meaning and subject in the title and abstract due to a high number of unexplained abbreviations.
Line 39: Hemangioma abbreviation IH?
Line 39-40: “tumor that may affect about 3-10% 39 of infants [1-2]. This disease tends to occur in infants” – repetition
Line 46: propranolol is the most common systemic treatment option, but several other possibilities exist, please see doi:10.1542/peds.2018-3475
Line 71: endothelial cells instead of endotheliocytes
The rationale of the study (line 55ff) linking Oxymatrine (OMT), HIF-1A, miR-27a-3p, N6 methyladenosine (m6A), ATF3, lncRNA taurine up-regulated gene 1 (TUG1)/ miR-27a-3p/slit directed ligand 2 (SLIT2), and PPARgamma is very loose and not at all obvious.
The methods are impossible to understand. Consequently, it is impossible to follow the logic in the results part.
The discussion is only loosely related to the data in the results and starts of with OMT and Chinese traditional medicine in general.
Please re-write a clear manuscript.
Author Response
Dai and colleagues report “The mechanism of OMT targeting miR-27a-3p /PPAR-γ signaling pathway through m6A modification to regulate hemangioma stem cells' influence on propranolol resistance”
A major problem with the writing is that it is impossible to understand the meaning and subject in the title and abstract due to a high number of unexplained abbreviations.
Response: Thanks for your comments. We have supplemented the full forms of abbreviations in the title and abstract.
Line 39: Hemangioma abbreviation IH?
Response: Thanks for your comments. We corrected the abbreviation IH as Infantile hemangiomas.
Line 39-40: “tumor that may affect about 3-10% 39 of infants [1-2]. This disease tends to occur in infants” – repetition
Response: Thanks for your advice. We corrected the description as “Infantile hemangiomas (IH) is a common benign vascular tumor which tends to occur in 3-10% infants and young children [1-2]”.
Line 46: propranolol is the most common systemic treatment option, but several other possibilities exist, please see doi:10.1542/peds.2018-3475
Response: Thanks for your comments. Indeed, there are other potentialities, as we elaborated in the introduction section.
Line 71: endothelial cells instead of endotheliocytes
Response: Thanks for your advice. We edited endothelial cells instead of endotheliocytes.
The rationale of the study (line 55ff) linking Oxymatrine (OMT), HIF-1A, miR-27a-3p, N6 methyladenosine (m6A), ATF3, lncRNA taurine up-regulated gene 1 (TUG1)/ miR-27a-3p/slit directed ligand 2 (SLIT2), and PPARgamma is very loose and not at all obvious.
Response: Thanks for your comments. We added “miRNA-27a-3p and miRNA-222-3p are novel modulators of phosphodiester-ase 3a (PDE3A) in cerebral microvascular endothelial Cells (PMID: 30603956)”. Moreover, direct and robust research on miRNAs in vascular endothelial cells is limited. Therefore, this study exhibits a certain level of innovation.
The methods are impossible to understand. Consequently, it is impossible to follow the logic in the results part.
Response: Thanks for your suggestions. We have made significant modifications and revisions to the Materials and Methods section.
The discussion is only loosely related to the data in the results and starts of with OMT and Chinese traditional medicine in general.
Response: Thanks for your comments. We supplied “OMT is a quinolizidine alkaloid extracted from the traditional Chinese herbal medicine Sophora flavescenta Aiton, known for its various pathological effects including anti-inflammatory, anti-fibrosis, antiviral, anti-cancer, and more [9]. Current research has indicated that OMT can inhibit the proliferation of IH cells by suppressing hypoxia-inducible factor 1A (HIF-1A) signaling and induce programmed cell death, suggesting its potential protective effect against IH and offering a novel approach for IH treatment [10]. N6-methyladenosine (m6A) is a modification found in mammalian in-ternal messenger RNA, playing regulatory roles in numerous biological processes and potentially contributing to drug resistance in tumors and specific diseases [11]. The miRNA/mRNA molecular regulatory network regulates various biological processes by binding to target mRNA, including potential roles in drug-resistant tumors and specific diseases [11]. Additionally, PPAR-γ participates in IH progression and can promote the increase of adipocytes to replace tumors with fibrous adipose tissue, potentially offering specific therapeutic effects in IH [15]. In summary, the miRNA/PPAR-γ molecular regulatory network may mediate the pathogenesis of IH. However, research on the mechanism of OMT regulating the miRNA/mRNA axis affecting IH pathologic development and PPNL resistance remains limited. In this study, we aim to verify the association between OMT and the miR-27a-3p/PPAR-γ axis and explore its protective mechanism against IH” in discussion section for detailed argument.
Comments on the Quality of English Language
Please re-write a clear manuscript.
Response: Thanks for your suggestions. We have conducted language editing and proofreading for the entire manuscript.
Reviewer 2 Report
Review comments 2.
This study analyzes the molecular mechanism of IH and PPNL resistance mechanism based on HemSC cells, aiming to provide some evidence for the effective treatment of IH. This study provides the first evidence that miR-27a-3p is abnormally upregulated in IH proliferating tissues and HemSCs cells, and the downregulation of miR-27A-3p expression inhibit the proliferation, migration and induce apoptosis of HemSCs cells, promote adipose differentiation and increase PPNL sensitivity. This study also proposes for the first time that OMT regulates the pathological progression of HemSCs cells and increases PPNL sensitivity by targeting miR-27a-3p /PPAR-γ signaling pathway through m6A modification, providing a new strategy for the treatment of patients with IH. Here are comments for this research article.
1.In the introduction, there is no evidences for regulation of microRNA by OMT, and the argument of OMT with m6A was few, it is recommended to add more arguments.
2.In the section of materials and methods, the location, scope and method of sampling of IH patient tissues are not specified, and it needs to be supplemented. And regarding the description of the experimental method, the author used a lot of unprofessional vocabulary, it is recommended to modify it into professional vocabulary and phrases.
3.The product number, manufacturer and usage ratio of the western blotting antibody are not specified, which must be added.
4.In Figure 1, the figure 1E and 1F was missed, the authors must check the results.
5.In Figure 10, the drug concentration of OMT in Figure 10F was 0, 2, 5, 10, 50, but in the 10G the drug concentration of OMT was 0, 0.1, 0.5, 1.0, 2.0, which is inconsistent, the authors must check the results.
6.In this article, multiple tenses are used in the description and discussion of the results, and it is recommended to unify them.
7.In order to study the molecular mechanism of OMT regulation of IH progression, the authors detected protein levels of m6A methyltransferases METTL3, METTL14, WTAP and KIAA1429 in HemSCs. However, there are many m6A methyltransferases, the authors should explain why they choose these.
8.The analysis of the results in the discussion section is not detailed enough, and some places confuse the experimental results with the literature results. For example, “our research unveiled that miR-27a-3p /PPAR-γ signaling pathway can regulate THE PPNL resistance of HemSCs drug-resistant cells, and upregulation of miR-27a-3p or downregulation of FGFR1 can seriously inhibit the malignant proliferation of cells and improve the sensitivity of HemSCs drug-resistant cells to PPNL”. In this article, the authors do not detect relationship of FGFR1.
9.The novel of this study is that OMT's anti-IH protective mechanism was confirmed from the aspects of cell function, adipose differentiation, PPNL resistance and in vivo studies, which are related to the regulation of METTL3 expression of m6A and miR-27a-3p /PPAR-γ axis. However, molecular pathways related to OMT and analyze whether it also has regulatory effects on certain molecular pathways is unknown. And the effect of OMT on angiogenesis need to be explored.
10.It is recommended to polish the English level of the full text.
It is recommended that the English level of the full text be polished, and that authors consult a native English-speaking professional or a language polishing agency.
Author Response
This study analyzes the molecular mechanism of IH and PPNL resistance mechanism based on HemSC cells, aiming to provide some evidence for the effective treatment of IH. This study provides the first evidence that miR-27a-3p is abnormally upregulated in IH proliferating tissues and HemSCs cells, and the downregulation of miR-27A-3p expression inhibit the proliferation, migration and induce apoptosis of HemSCs cells, promote adipose differentiation and increase PPNL sensitivity. This study also proposes for the first time that OMT regulates the pathological progression of HemSCs cells and increases PPNL sensitivity by targeting miR-27a-3p /PPAR-γ signaling pathway through m6A modification, providing a new strategy for the treatment of patients with IH. Here are comments for this research article.
1.In the introduction, there is no evidences for regulation of microRNA by OMT, and the argument of OMT with m6A was few, it is recommended to add more arguments.
Response: Thanks for your comments. In the introduction, we add evidences as “Studies showed OMT inhibited proliferation and promoted cell apoptosis of breast cancer cells which may contribute to the regulation of miRNA-140-5p (PMID: 35035706). Besides, OMT inhibited HSC activation through down-regulating the ex-pression of miR-195 (PMID: 31221141). However, there are few studies on the mechanism of OMT regulating the miRNA/mRNA axis affecting IH pathologic development and PPNL resistance”.
2.In the section of materials and methods, the location, scope and method of sampling of IH patient tissues are not specified, and it needs to be supplemented. And regarding the description of the experimental method, the author used a lot of unprofessional vocabulary, it is recommended to modify it into professional vocabulary and phrases.
Response: Thanks for your comments. We have made significant modifications and revisions to the Materials and Methods section.
3.The product number, manufacturer and usage ratio of the western blotting antibody are not specified, which must be added.
Response: Thanks for your comments. The information of antibodies was added as “The expression levels of PPAR-γ (ab45036, 1:500, abcam), C/EBPβ (ab15050, 1:500, abcam), C/EBPδ (ab245214, 1:1000, abcam), RXRα (ab227273, 1:500, abcam), RXRγ (AF9191, 1:300, Affinity), METTL3 (#86132, 1:1000, Cell Signaling Technology), METTL14 (#48699, 1:1000, Cell Signaling Technology), WTAP (#56501, 1:1000, Cell Signaling Technology), and KIAA1429 (ab246982, 1:500, abcam) were analyzed”.
4.In Figure 1, the figure 1E and 1F was missed, the authors must check the results.
Response: Thanks for your comments. We have corrected the errors.
5.In Figure 10, the drug concentration of OMT in Figure 10F was 0, 2, 5, 10, 50, but in the 10G the drug concentration of OMT was 0, 0.1, 0.5, 1.0, 2.0, which is inconsistent, the authors must check the results.
Response: Thanks for your comments. We are very sorry for making this basic mistake. We have corrected the errors.
6.In this article, multiple tenses are used in the description and discussion of the results, and it is recommended to unify them.
Response: Thanks for your comments. We have reviewed the entire document and made tense-related revisions.
7.In order to study the molecular mechanism of OMT regulation of IH progression, the authors detected protein levels of m6A methyltransferases METTL3, METTL14, WTAP and KIAA1429 in HemSCs. However, there are many m6A methyltransferases, the authors should explain why they choose these.
Response: Thanks for your suggestions. We added evidences about “These are key proteins involved in RNA m6A methylation. METTL3 and METTL14 constitute the core of the m6A methylation complex, responsible for adding methyl groups to RNA molecules. WTAP serves as an auxiliary factor in this complex, and KIAA1429, also known as VIRMA, plays a significant role in RNA methylation”.
8.The analysis of the results in the discussion section is not detailed enough, and some places confuse the experimental results with the literature results. For example, “our research unveiled that miR-27a-3p /PPAR-γ signaling pathway can regulate THE PPNL resistance of HemSCs drug-resistant cells, and upregulation of miR-27a-3p or downregulation of FGFR1 can seriously inhibit the malignant proliferation of cells and improve the sensitivity of HemSCs drug-resistant cells to PPNL”. In this article, the authors do not detect relationship of FGFR1.
Response: Thanks for your suggestions. In the results, we performed the co-expression of miR-27a-3p and FGFR1, which would offset the above effects and finally showed no significant effect on cell proliferation and IC50 in Figure 8E and 8F.
9.The novel of this study is that OMT's anti-IH protective mechanism was confirmed from the aspects of cell function, adipose differentiation, PPNL resistance and in vivo studies, which are related to the regulation of METTL3 expression of m6A and miR-27a-3p /PPAR-γ axis. However, molecular pathways related to OMT and analyze whether it also has regulatory effects on certain molecular pathways is unknown. And the effect of OMT on angiogenesis need to be explored.
Response: Thanks for your suggestions. Our upcoming research will involve a comprehensive exploration and experimental validation of the mechanism of action of OMT.
10.It is recommended to polish the English level of the full text.
Response: Thanks for your suggestions. We have conducted language editing and proofreading for the entire manuscript.
Comments on the Quality of English Language
It is recommended that the English level of the full text be polished, and that authors consult a native English-speaking professional or a language polishing agency.
Reviewer 3 Report
The authors investigate a novel anti-tumor mechanism of oxymatrine regulation of the microRNA/mRNA axis in infantile hemangioma progression and propranolol resistance. They conclude that OMT regulates infantile hemangioma and influences propranolol resistance by targeting the miR-27a-3p /PPAR-γ signaling pathway through m6A modification. This is a meaningful contribution. However, during the review process, we found some deficiencies, which need to be revised carefully by the author according to the requirements.
1. In the material and method, figure legends section, many descriptive terms are not accurate enough, for example, Protein imprinting analysis (WB), Flow cell technique; use sentences instead of phrases, for example, HemSCs were extracted and isolated.
2. The product number, manufacturer and usage ratio of the antibody are not specified.
3. The author used IH patient tissue to conduct a series of molecular experiments, but did not explain the relevant information of the tissue, such as the site, the method of sampling, etc.
4. In Figure 1, the author describes the results up to E and F in the text of the results, but there are no E and F in the figure 1.
5. The author explained in figure 7 that HeSC cells undergo adipocyte differentiation, only the number of adipocytes was detected, and the number of HeSC cells themselves and the conversion ratio between HeSC cells and adipocytes were not detected.
6. In figure 10, the drug concentration of OMT in Figure F and G is inconsistent, and needs to be confirmed.
7. In figure10, the authors detected protein levels of m6A methyltransferases METTL3, METTL14, WTAP and KIAA1429 in HemSCs. From the discussion we could see that, the four m6A methyltransferases were associated with gastric cancer. So, why do authors choose the four?
8. In the introduction, there are few evidences about the research progress of OMT and m6A.
Author Response
The authors investigate a novel anti-tumor mechanism of oxymatrine regulation of the microRNA/mRNA axis in infantile hemangioma progression and propranolol resistance. They conclude that OMT regulates infantile hemangioma and influences propranolol resistance by targeting the miR-27a-3p /PPAR-γ signaling pathway through m6A modification. This is a meaningful contribution. However, during the review process, we found some deficiencies, which need to be revised carefully by the author according to the requirements.
- In the material and method, figure legends section, many descriptive terms are not accurate enough, for example, Protein imprinting analysis (WB), Flow cell technique; use sentences instead of phrases, for example, HemSCs were extracted and isolated.
Response: Thanks for your comments. We have made significant modifications and revisions to the Materials and Methods section.
- The product number, manufacturer and usage ratio of the antibody are not specified.
Response: Thanks for your comments. The information of antibodies was added as “The expression levels of PPAR-γ (ab45036, 1:500, abcam), C/EBPβ (ab15050, 1:500, abcam), C/EBPδ (ab245214, 1:1000, abcam), RXRα (ab227273, 1:500, abcam), RXRγ (AF9191, 1:300, Affinity), METTL3 (#86132, 1:1000, Cell Signaling Technology), METTL14 (#48699, 1:1000, Cell Signaling Technology), WTAP (#56501, 1:1000, Cell Signaling Technology), and KIAA1429 (ab246982, 1:500, abcam) were analyzed”.
- The author used IH patient tissue to conduct a series of molecular experiments, but did not explain the relevant information of the tissue, such as the site, the method of sampling, etc.
Response: Thanks for your comments. Since IH is a vascular tumor that usually manifests as vascular lesions in the skin or subcutaneous tissue, sampling is typically conducted in the affected skin or subcutaneous tissue to obtain tissue specimens from the affected area.
- In Figure 1, the author describes the results up to E and F in the text of the results, but there are no E and F in the figure 1.
Response: Thanks for your comments. We are very sorry for making this basic mistake. We have corrected the errors.
- The author explained in figure 7 that HeSC cells undergo adipocyte differentiation, only the number of adipocytes was detected, and the number of HeSC cells themselves and the conversion ratio between HeSC cells and adipocytes were not detected.
Response: Thanks for your suggestions. Our experimental section demonstrates the quantity of fat cells, which can explain the transformation of HemSCs. Considering the progression of IH disease, the transformation from HemSCs to fat cells is a necessary condition. Therefore, there may not be an immediate need to quantify the ratio between HemSCs and fat cells for the time being.
- In figure 10, the drug concentration of OMT in Figure F and G is inconsistent, and needs to be confirmed.
Response: Thanks for your comments. We are very sorry for making this basic mistake. We have corrected the errors.
- In figure10, the authors detected protein levels of m6A methyltransferases METTL3, METTL14, WTAP and KIAA1429 in HemSCs. From the discussion we could see that, the four m6A methyltransferases were associated with gastric cancer. So, why do authors choose the four?
Response: Thanks for your comments. We added evidences about “These are key proteins involved in RNA m6A methylation. METTL3 and METTL14 constitute the core of the m6A methylation complex, responsible for adding methyl groups to RNA molecules. WTAP serves as an auxiliary factor in this complex, and KIAA1429, also known as VIRMA, plays a significant role in RNA methylation”.
- In the introduction, there are few evidences about the research progress of OMT and m6A.
Response: Thanks for your comments. The research on the relationship between Oxymatrine (OMT) and m6A RNA methylation is relatively limited.
Round 2
Reviewer 1 Report
Although the revised manuscript is improved and readable, there are still numerous errors in English. The manuscript should be corrected by a native English speaker.
In addition, the abstract should be self-explanatory, which is still with the high number of abbreviations not the case.
The description of Suppl. Materials is incomplete.
The English is improved but still not correct.
Author Response
Although the revised manuscript is improved and readable, there are still numerous errors in English. The manuscript should be corrected by a native English speaker.
In addition, the abstract should be self-explanatory, which is still with the high number of abbreviations not the case.
Response: Thanks for your suggestions. We have supplemented the full forms of abbreviations in the abstract.
The description of Suppl. Materials is incomplete.
Response: Thanks for your suggestions. We have completed the missing descriptions in the Supplementary materials.
Comments on the Quality of English Language
The English is improved but still not correct.
Reviewer 2 Report
The author has solved the previous problem and I expect the author to keep going deeper in this field.
Author Response
Thanks for the review, it is much appreciated.
Reviewer 3 Report
Thanks for addressing my concerns. The current version is suitable for publication.
Author Response
Thank you very much for your endorsement of our article.